# Hypoxic microenvironment shapes HIV-1 replication and latency

Xiaodong Zhuang[1,4], Isabela Pedroza-Pacheco[1,4], Isabel Nawroth[2], Anna E. Kliszczak[1], Andrea Magri[1], Wayne Paes [1], Claudia Orbegozo Rubio[1], Hongbing Yang[1], Margaret Ashcroft [3], David Mole[1], Peter Balfe [2], Persephone Borrow [1,5] & Jane A. McKeating [1,5 ✉]

Viral replication is defined by the cellular microenvironment and one key factor is local oxygen tension, where hypoxia inducible factors (HIFs) regulate the cellular response to oxygen. Human immunodeficiency virus (HIV) infected cells within secondary lymphoid tissues exist in a low-oxygen or hypoxic environment in vivo. However, the majority of studies on HIV replication and latency are performed under laboratory conditions where HIFs are inactive. We show a role for HIF-2α in restricting HIV transcription via direct binding to the viral promoter. Hypoxia reduced tumor necrosis factor or histone deacetylase inhibitor, Romidepsin, mediated reactivation of HIV and inhibiting HIF signaling-pathways reversed this phenotype. Our data support a model where the low-oxygen environment of the lymph node may suppress HIV replication and promote latency. We identify a mechanism that may contribute to the limited efficacy of latency reversing agents in reactivating HIV and suggest new strategies to control latent HIV-1.

[1] Nuffield Department of Clinical Medicine, University of Oxford, Oxford OX3 7FZ, UK. [2] Institute of Immunity and Immunotherapy, University of Birmingham, Birmingham B15 2TT, UK. [3] Department of Medicine, University of Cambridge, Cambridge Biomedical Campus, Cambridge CB2 0AH, UK. [4] These authors contributed equally: Xiaodong Zhuang, Isabela Pedroza-Pacheco. [5] These authors jointly supervised this work: Persephone Borrow, Jane A McKeating. ✉email: jane.mckeating@ndm.ox.ac.uk

Viral replication in host cells is shaped by the cellular microenvironment. One important environmental factor to consider is the local oxygen ($O_2$) tension, which can vary widely depending on the metabolic demand and blood supply of the tissue or organ[1,2]. While most in vitro studies on viral replication utilize cells cultured at atmospheric oxygen levels (~20% $O_2$), the vast majority of human tissues have oxygen levels much lower than this (referred to as hypoxia). Hypoxia can have opposing effects on viral proliferation, inducing the replication of Epstein–Barr virus[3–5], Kaposi sarcoma-associated herpesvirus[4,6], human T-cell leukemia virus[7], and human polyomavirus BK[8], while suppressing the infectivity of adenovirus[9] and Moloney murine leukemia virus[10]. These contrasting effects are likely to reflect the variable oxygen tension at the sites of virus replication and the complex interplay between viruses and their hosts[11].

Mammalian cells adapt to low oxygen through an orchestrated transcriptional response regulated by hypoxia inducible factors (HIFs) that bind specific motifs or hypoxia response elements (HREs) in the promoter/enhancer elements of their target genes[12,13]. HIFs are heterodimeric transcription factors comprising a HIFα subunit (HIF-1α or HIF-2α) and a HIF-1β subunit and are regulated by oxygen-dependent and independent stress signals. HIFs control a wide range of genes involved in many cellular processes including energy metabolism and inflammation[14,15]. Mounting evidence reveals a role for HIFs in a number of diseases including cancer and inflammatory conditions, where pharmacological approaches to modulate HIF activity offer promising therapeutic avenues[16,17]. However, the role of HIFs in chronic viral infection are not well understood.

Human immunodeficiency virus type 1 (HIV-1) is not effectively contained by the host immune response. Although antiretroviral therapies (ARTs) control viral replication they fail to eradicate integrated copies of the viral genome that constitute a long-lived reservoir of infection. HIV-1 primarily replicates in CD4 T cells and the major reservoirs are thought to reside in lymphoid tissues[18–20], which contain quiescent or latently-infected cells that are not sensitive to ART[21–24]. Other cell populations of lymphoid, myeloid, and stromal origins may contribute to the HIV reservoir; however, their role in HIV-1 persistence is less well-understood, in part due to the challenges of sampling tissue at multiple sites. Viral latency is considered a reversible state of nonproductive infection that evades detection by host immune responses. 'Kick and kill' strategies are being developed to combat HIV persistence by activating quiescent HIV with latency-reversing agents to enable immune mediated clearance of infected cell reservoirs[25–27].

Lymphoid organs operate at oxygen levels in the range of 0.5–4.5% $O_2$[28–30], which are sufficient to activate the HIF pathway[31]. However, the majority of in vitro studies investigating HIV replication and latency have been performed at 20% $O_2$ where HIFs are usually inactive, so roles played by HIFs in regulating HIV replication may have been overlooked. In this study we evaluated the effect of low oxygen and HIF signalling on HIV-1 replication in primary CD4 T cells and latency models. We uncover a role for HIF-2α (the HIF-2α/HIF-1ß dimer) in repressing HIV transcription through direct interaction with a conserved HRE in the viral promoter. Furthermore, we show that a low-oxygen environment reduces the reactivation of latent HIV. Our findings highlight the importance of oxygen availability in regulating HIV replication and identify a mechanism that may contribute to the limited efficacy of current latency-reversing agents to reactivate HIV in vivo.

## Results

**Low oxygen reduces HIV-1 replication in CD4 T cells.** HIV-1 replicates preferentially in activated CD4 T cells[32] and since HIFs regulate T-cell responses to activating stimuli[33] we elected to study the effect of hypoxia on HIV replication in preactivated CD4 T cells. Peripheral blood mononuclear cells (PBMCs) from HIV-seronegative donors were depleted of CD8 T cells, activated with antibodies to CD3/CD28 for 3 days and infected with HIV-1 (NL4.3-Bal). Unbound virus was removed by washing and the cells cultured in 1% $O_2$, an oxygen tension previously employed to study T-cell effector responses[34,35], or standard 'normoxic' laboratory conditions of 20% $O_2$ for 4 days (Fig. 1a). We selected to study HIV replication during the exponential phase of virus expansion (Supplementary Fig. 1a). The low-oxygen conditions had no adverse effects on T-cell viability and did not reduce the activation status of the cells: T-cell activation was in fact modestly enhanced, as evidenced by an increase in the expression of PD-1, CD25, and CD38 (Supplementary Fig. 2). However, we observed a significant reduction in HIV replication assessed by measuring extracellular HIV p24 core antigen levels at days 2 and 4 post-infection in the low-oxygen cultures (Fig. 1b, c, Supplementary Fig. 1b–d).

To confirm that the CD4 T cells had responded to the low-oxygen conditions we measured expression of the HIF-regulated gene *SLC2A1* that encodes Glucose transporter 1 (GLUT-1). We observed an increase in GLUT-1 protein and mRNA levels in cells cultured under 1% $O_2$ (Fig. 1d). Since CD4 T cells can migrate between sites of variable oxygen tension in vivo we were interested to investigate the effect of reoxygenation on HIV replication.

Oxygen reperfusion of hypoxic cells results in a rapid and time-dependent loss of HIFs. Transferring infected cultures after 2 days at 1% $O_2$ to 20% $O_2$ had a minimal effect on p24 antigen expression (Fig. 1c), demonstrating a continued impact of low-oxygen-induced repression of HIV replication over the 48 h reoxygenation time period. In contrast, after 2 days of reoxygenation both GLUT-1 protein and mRNA levels returned to those observed in normoxic cultures (Fig. 1d). We noted some interdonor variability in the low-oxygen-dependent inhibition of HIV replication and GLUT-1 expression; however, there was no significant association between these parameters (Supplementary Fig. 1e). Together, these results show that a low-oxygen environment suppresses HIV-1 replication in activated CD4 T cells.

**Low-oxygen regulates postintegration of HIV replication.** The cellular response to low oxygen includes the transcriptional activation of an array of host genes involved in cellular proliferation, differentiation, and energy metabolism that could influence multiple aspects of the HIV life cycle. The first steps in HIV infection are dependent on the expression of CD4 and CCR5 or CXCR4, cell-surface receptors required for viral binding and internalization. Culturing activated CD4 T cells under 20% or 1% $O_2$ conditions did not reduce the frequency of positive cells (%) or expression levels (geometric Mean Fluorescence Intensity; gMFI) of these entry receptors: CXCR4 and CD4 expression was increased under 1% $O_2$ conditions (Fig. 2a, b), suggesting that low oxygen is unlikely to reduce HIV entry. Following HIV entry into a target cell the encapsidated RNA genome is reverse transcribed and the capsid traffics to the nucleus where the newly synthesized proviral DNA integrates into host chromosomal DNA. To investigate whether low oxygen regulates any of these early steps in the viral life cycle we measured the earliest product of HIV reverse transcription, a short 197 bp product defined as 'strong stop', along with the secondary 1st strand and 2nd strand transfer cDNA products, in HIV-1 infected CD4 T cells cultured at 20% or 1% $O_2$ for 2 days. Low oxygen had no significant effect on these viral

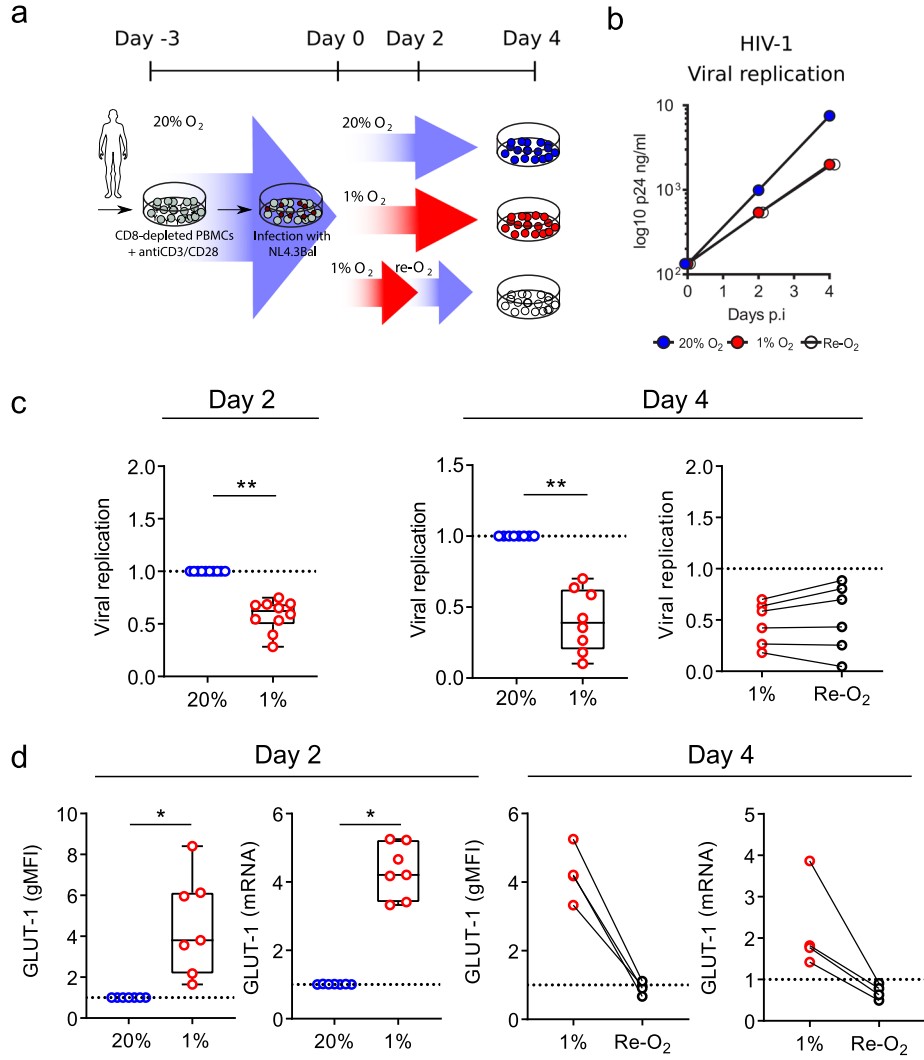

**Fig. 1 Low oxygen reduces HIV replication. a** Diagram depicting the protocol used to study the effect of low oxygen on HIV-1 replication. CD8-depleted PBMC activated in 20% $O_2$ were infected with HIV-1 NL4.3-Bal and cultured under 20% or 1% $O_2$ conditions, and viral replication was assessed at 2 and 4 days post-infection by measuring extracellular p24 antigen levels. In some experiments, cells under 1% $O_2$ were reoxygenated for the last 2 days of culture (Re-$O_2$). **b** Representative example of HIV replication in PBMC from a single donor cultured under various oxygen conditions. **c** HIV replication in PBMC from multiple donors. Extracellular p24 antigen levels in cultures from each donor are expressed relative to the normoxic controls. Each symbol represents data from an individual donor where two independent biological replicates were analysed. Group medians, range and quartiles are shown in the left and centre panels and paired samples are indicated in the right panel ($n = 6$–10, mean ± SEM, Wilcoxon matched-pairs signed rank test). **d** HIF target gene GLUT-1 protein (gMFI) and mRNA levels were measured by flow cytometry and quantitative RT-PCR, respectively in HIV-infected cells cultured under the conditions depicted in panel **a**. GLUT-1 expression under normoxic conditions is expressed as the gMFI or mRNA relative to values in 20% $O_2$ at the timepoint indicated. Each symbol represents data from an individual donor. In the panels on the left, group medians, range, and quartiles are shown, and paired samples are indicated in the right panel ($n = 4$–8; mean ± SEM, Wilcoxon matched-pairs signed rank test).

parameters (Fig. 2c). The frequency of HIV integration events was also comparable under 20% or 1% $O_2$ conditions (Fig. 2d). Importantly, we observed a significant reduction in intracellular HIV RNA levels in the low-oxygen-infected cultures (Fig. 2e), consistent with a role for oxygen tension in regulating HIV transcriptional activity.

**Low oxygen limits HIV promoter activity via HIF.** To evaluate the effect of low oxygen on HIV promoter activity we used the Jurkat 1G5 and Hela TZM-bl cell lines that bear integrated copies of the luciferase gene under the control of the HIV-1 long terminal repeat (LTR)[36]. These reporter lines are commonly used to monitor HIV replication: following infection the virus encoded transcriptional activator, Tat, binds the LTR and induces

luciferase expression. Low oxygen significantly reduced HIV-1 (NL4.3-Bal) dependent LTR activation in 1G5 cells (Fig. 3a). Since TZM-bl cells are engineered to express CD4, CCR5, and CXCR4 this enabled us to compare the effect of low oxygen on the CXCR4-tropic strain of HIV NL4.3 with the chimeric NL4.3-Bal clone that encodes a CCR5-tropic envelope glycoprotein. Low oxygen repressed the replication of both HIV strains, demonstrating that the low-oxygen phenotype is independent of co-receptor usage (Fig. 3a). We confirmed that low-oxygen conditions had no significant effect on the early steps of viral life cycle (strong stop, 1st and 2nd strand products) and HIV integrated copies in both 1G5 and TZM-bl cells (Supplementary Fig. 3). TZM-bl cells showed low basal LTR activity and transient expression of TAT induced a 7-fold increase in promoter activity (Supplementary Fig. 4a): low oxygen repressed basal and

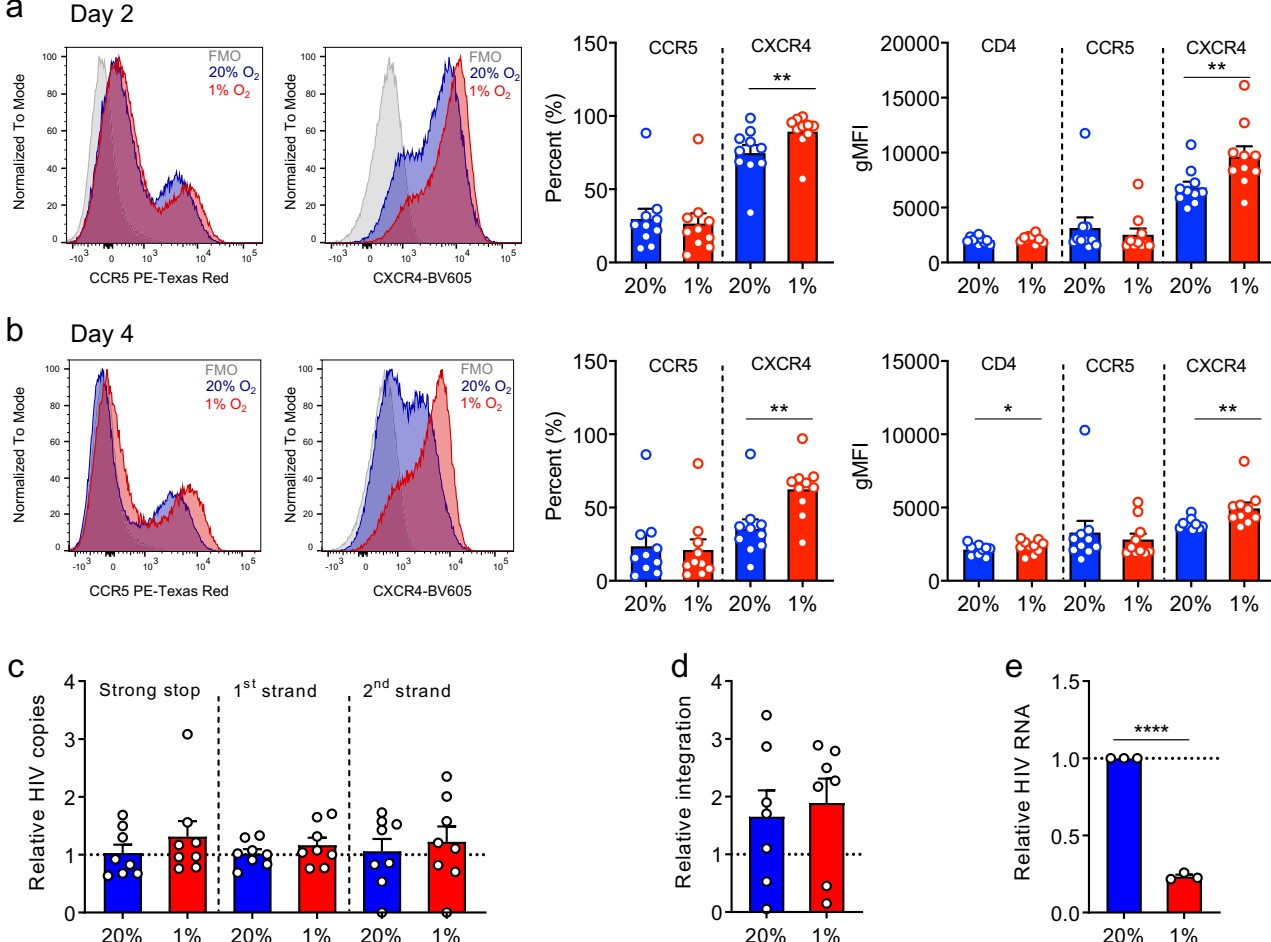

**Fig. 2 Oxygen regulates postintegration steps of HIV life cycle. a, b** Activated CD8-depleted PBMC were cultured under 20% or 1% $O_2$ for 2 (**a**) or 4 (**b**) days and surface expression of CD4, CCR5, and CXCR4 was measured by flow cytometry, setting gates for each marker based on fluorescence minus one (FMO) controls. The percentage of positive cells and their mean level of expression (geometric mean fluorescence intensity, gMFI) of each receptor were assessed within live CD3 + CD4 + T cells. Histogram plots illustrating the staining of CD4 T cells from one representative donor are shown on the left and summary plots of data from 10 donors are shown on the right (mean + SEM, Wilcoxon matched-pairs signed rank test). **c, d** Activated CD4 T cells isolated from human PBMCs were infected with HIV NL4.3-Bal under 20% or 1% $O_2$ for 2 days. HIV strong stop, 1st and 2nd strand products along with the number of HIV integrated copies in the hypoxic cultures are expressed relative to the normoxic controls. Each symbol represents data from an individual donor ($n = 7$–8; mean ± SEM, Wilcoxon matched-pairs signed rank test). **e** Activated CD4 T cells isolated from human PBMCs were infected with HIV NL4.3-Bal under 20% or 1% $O_2$ for 2 days. Cellular HIV RNA levels were measured and the data expressed relative to 20% cultures. Each symbol represents data from an individual donor ($n = 3$; mean ± SEM, Wilcoxon matched-pairs signed rank test).

TAT-activated LTR activity to comparable levels (Fig. 3b). To extend these studies to more physiologically relevant cell types we transiently expressed the HIV-LTR-Luc reporter (strain LAI) in Jurkat and primary CD4 T cells and showed that low oxygen reduced promoter activity in both (Fig. 3c). We next examined the role of NF-kB, a major host factor that promotes HIV transcription, in defining the LTR response to low oxygen. Luciferase expression was repressed in Jurkat cells expressing either a wild-type (WT) or delta NF-kB LTR-Luc reporter under low-oxygen conditions (Fig. 3d). To investigate whether low-oxygen-mediated repression of the HIV-LTR was conserved in diverse HIV subtypes we transfected Jurkat cells with a panel of LTR-Luc reporter plasmids encoding promoter regions cloned from diverse HIV-1 clades and as a control we transfected cells with a HRE-Luc reporter. Low oxygen activated the HRE-dependent reporter 3-fold (Supplementary Fig. 4b) as expected and repressed the activity of all HIV LTRs, showing a pan-genotypic effect (Fig. 3e).

The cellular response to low oxygen is regulated by three HIF transcription factors (HIF-1, HIF-2, and HIF-3), each comprising a heterodimer of a separate alpha and a common beta subunit. The best understood, HIF-1 and HIF-2, are regulated via an oxygen-dependent degradation domain in their alpha subunit, whereas the beta subunit is constitutively expressed. Since HIF-1α and HIF-2α can show nonoverlapping and sometimes opposing functions[14] we were interested to investigate their individual roles in regulating the HIV promoter. By performing transient siRNA silencing of HIF-1α or HIF-2α in Jurkat T cells prior to culturing under 20% or 1% $O_2$ conditions we were able to reduce expression of each HIF isoform under low-oxygen conditions (Fig. 4a). To assess how knockdown (KD) of each HIF-α isoform impacted on the low-oxygen-mediated repression of HIV-LTR activity we codelivered each of the siRNAs into Jurkat cells with a HIV-LTR-Luc or control HRE-Luc reporter. Low oxygen still reduced HIV-LTR activity in HIF-1α KD cells but this phenotype was ablated in the HIF-2α KD cells (Fig. 4a), supporting a role for HIF-2α in regulating HIV promoter activity. As expected, low oxygen activated the HRE reporter and HIF-1α KD ablated this effect. In contrast, HIF-2α KD had a minimal effect on HRE

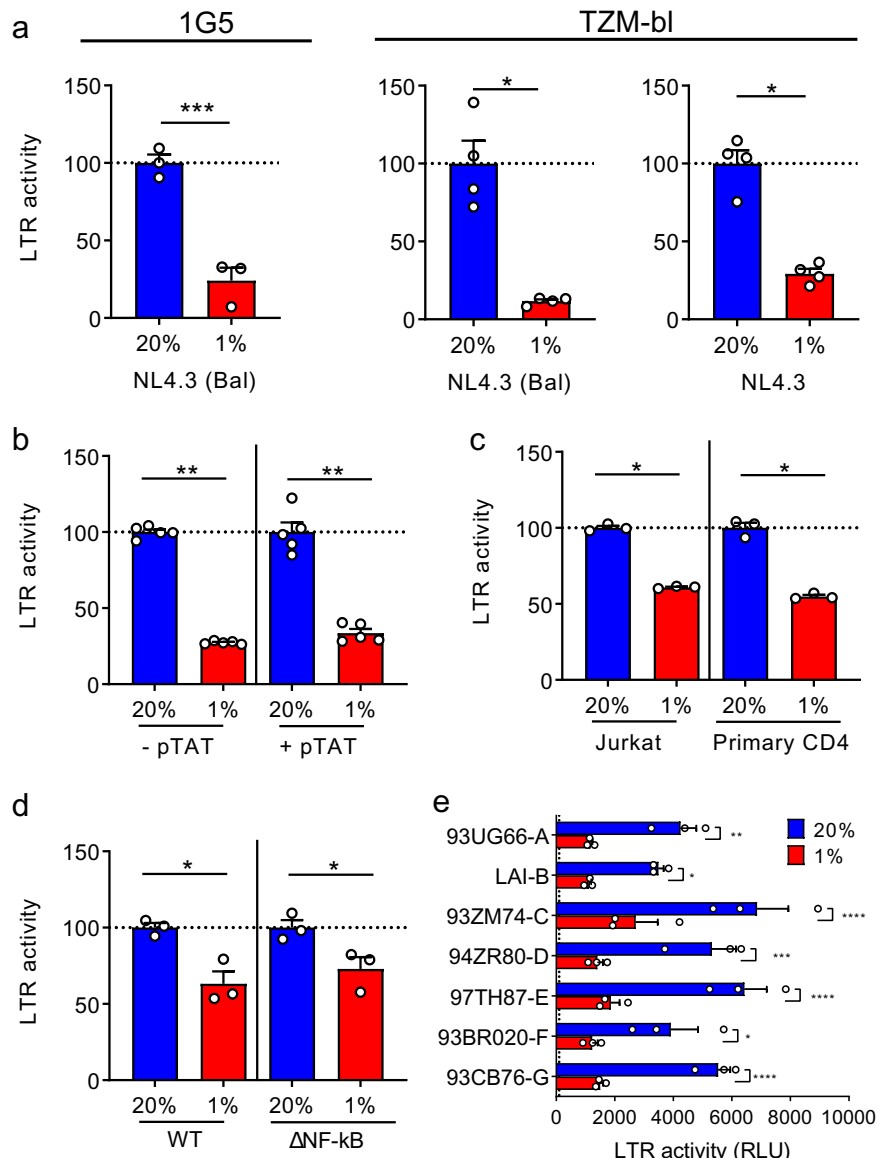

**Fig. 3 Low oxygen reduces HIV transcription. a** 1G5 or TZM-bl cells were infected with HIV NL4.3-Bal (CCR5-tropic) or NL4.3 (CXCR4-tropic) and cultured at 20% or 1% $O_2$ for 48 h. HIV-dependent LTR activation was evaluated by measuring the net luciferase activity in infected cultures and the data expressed relative to the normoxic controls ($n = 3$–4, mean ± SEM, Mann–Whitney analysis). **b** TZM-bl cells were transfected with pTAT and cultured under 20% or 1% $O_2$ conditions. LTR activity was assessed 48 h later by evaluating luciferase expression in cell lysates and the data expressed relative to the normoxic controls ($n = 5$, mean ± SEM, Mann–Whitney analysis). **c** Jurkat cells or primary CD4 T cells were infected with VSV-G-pseudotyped HIV NL4.3-Luc and cultured at 20% or 1% $O_2$ for 48 h. Luciferase activity is expressed relative to values in the normoxic conditions ($n = 3$, mean ± SEM, Mann–Whitney analysis). **d** Jurkat cells transfected with WT or NF-kB motif deleted HIV-LTR-Luc reporter were cultured at 20% or 1% $O_2$ for 48 h and LTR activity assessed by evaluating luciferase expression in cell lysates. The data are expressed relative to the normoxic conditions ($n = 3$, mean ± SEM, Mann–Whitney analysis). **e** HIV subtype LTR-Luc reporters were transfected into Jurkat cells and cultured in 20% or 1% $O_2$ conditions for 48 h. Luciferase values (relative light units, RLU) are reported, and the dashed line represents the mean luciferase value of nontransfected cells cultured at 20% and 1% $O_2$ ($n = 3$, mean ± SEM, Two-way ANOVA analysis).

activity (Fig. 4a), suggesting that the HRE reporter is primarily regulated via HIF-1α in this T-cell line. To extend these observations we assessed the ability of the pharmacological HIF pathway inhibitor NSC-134754 to regulate HIF expression in Jurkat T cells. NSC-134754 showed a dose-dependent reduction in HIF-1α and HIF-2α expression in hypoxic T cells (Fig. 4b). We confirmed that pharmacological inhibition of HIF signaling with NSC-134754 or silencing HIF-1α or HIF-2α had no effect on Jurkat T-cell viability (Supplementary Fig. 5a). We selected the most genetically different members of our HIV-LTR-Luc reporter panel (LAI—clade B, 93ZM74—clade C and 97TH87-2 clade E)

and found that treating Jurkat cells with NSC-134754 reversed the inhibitory effect of low oxygen on all three of these HIV promoters (Fig. 4c). Furthermore, NSC-134754 ablated the effects of low oxygen on HIV-LTR LAI activity in primary CD4 T cells (Fig. 4d) without impairing their viability (Supplementary Fig. 5b). These studies highlight a role for HIFs in repressing HIV promoter activity under low oxygen.

An important family of enzymes that catalyze the hydroxylation and demethylation of DNA and RNA known as the 2-oxoglutarate (2-OG) oxygenases require oxygen, Fe(II) and 2-oxoglutarate (2-OG) for their activity[37]. Dimethyloxalylglycine

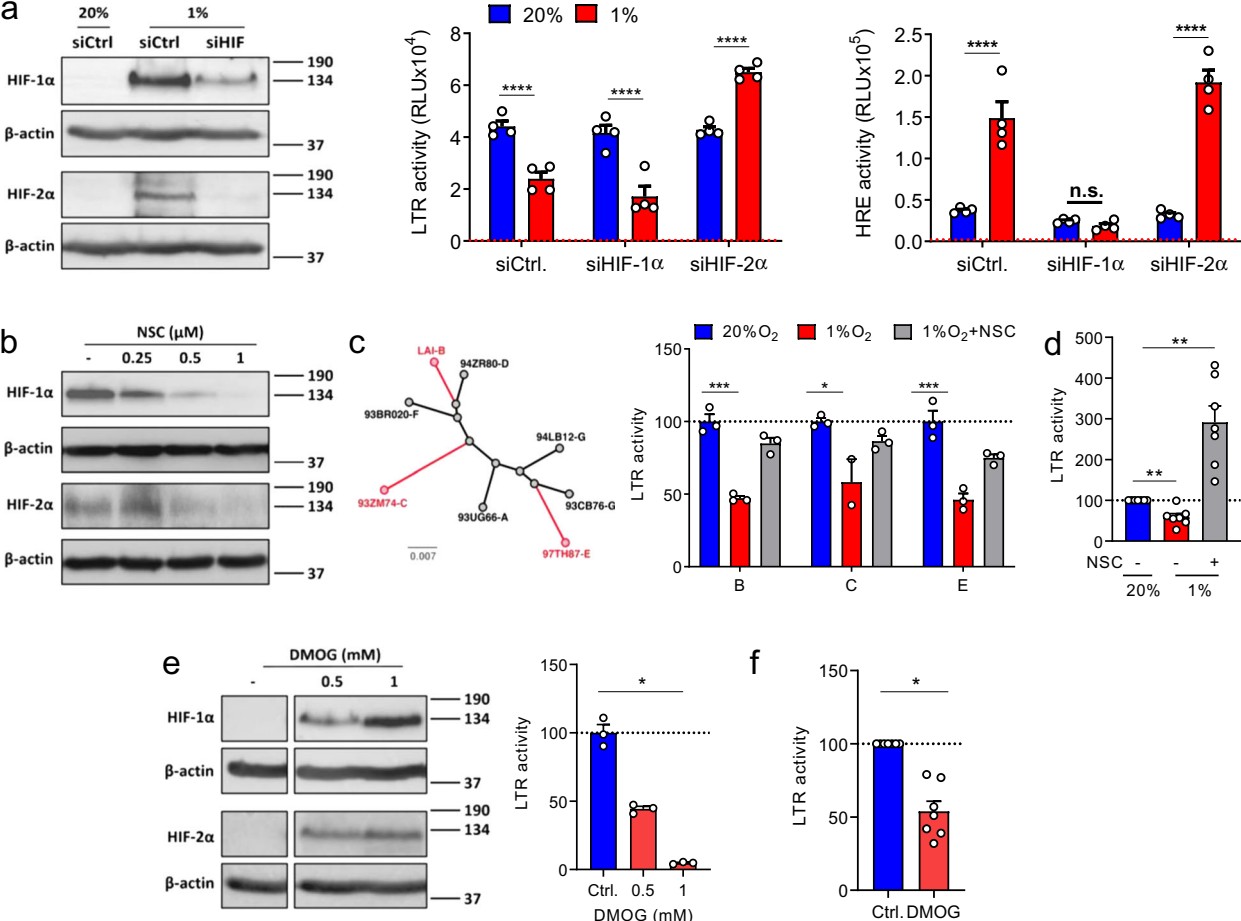

**Fig. 4 Silencing and pharmacological inhibition of HIFs restores HIV-LTR promoter activity under low oxygen. a** Jurkat cells were co-transfected with HIV-LTR-Luc or HRE-Luc along with siRNAs targeting HIF-1α and HIF-2α. 24 h later, transfected cells were cultured under 20% or 1% $O_2$ conditions for 24 h and LTR or HRE activity assessed by luciferase assay. HIF expression was assessed by western blotting. The middle and right panels show Luc expression (Relative light units (RLU) and the red the dashed line represents the mean RLU of nontransfected cells ($n = 4$, mean ± SEM, Two-way ANOVA analysis). **b** Jurkat cells were cultured under 1% $O_2$ for 24 h in the presence of increasing concentrations of the HIF pathway inhibitor NSC-134754 and assessed for HIF-1α and HIF-2α expression by western blotting. **c** HIV-LTR sequences were aligned and neighbor joining trees constructed (bootstrap = 1000) using ClustalX. Jurkat cells were transfected with selected HIV subtype LTR-Luc reporters and treated with NSC-134754 under 1% $O_2$ for 24 h. LTR activity was determined by luciferase assay and the data expressed relative to the normoxic conditions ($n = 3$, mean ± SEM, Two-way ANOVA analysis). **d** Activated CD4 T cells isolated from human PBMCs were infected with VSV-G-pseudotyped HIV NL4.3-Luc and treated with NSC-134754 under 1% $O_2$ for 24 h. LTR activity was determined by luciferase assay and the data expressed relative to the normoxic conditions. Each symbol represents data from an individual donor ($n = 7$; mean ± SEM, Wilcoxon matched-pairs signed rank test). **e** VSV-G-pseudotyped HIV NL4.3-Luc infected Jurkat cells were treated with DMOG under 20% $O_2$ conditions for 24 h and luciferase activity measured. Data are expressed relative to the control untreated cells ($n = 3$, mean ± SEM, One-way ANOVA analysis). Cell lysates were collected to probe for HIF-1α and HIF-2α expression by western blotting. **f** VSV-G-pseudotyped HIV NL4.3-Luc infected activated CD4 T cells were treated with DMOG under 20% $O_2$ conditions for 24 h and luciferase activity measured. Data are expressed relative to the control untreated cells. Each symbol represents data from an individual donor ($n = 7$; mean ± SEM, Wilcoxon matched-pairs signed rank test).

(DMOG) is a broad-spectrum inhibitor of the 2-OG oxygenases and is widely used as a hypoxia mimetic. DMOG stabilized HIF-1α and HIF-2α in Jurkat cells and significantly reduced HIV-1 LTR activity in a dose-dependent manner (Fig. 4e) without impairing Jurkat cell viability (Supplementary Fig. 5a). DMOG induced a similar inhibition of LTR activity in primary CD4 T cells (Fig. 4f).

HIFs bind a conserved motif (RCGTGC) defined as the HRE in the promoter and enhancer regions of responsive genes. Screening HIV sequences deposited in the Los Alamos Database revealed that approximately one third (330/897 sequences) encode an antisense HRE at position 293-287 in the U3 region of the LTR (Fig. 5a). All of the HIV subtype LTR-Luc reporters used in the experiments reported in Figs. 3 and 4 encode this

motif (Fig. 5b). To ascertain whether the HRE in the HIV promoter defined the response to low oxygen we generated a HIV-LTR-Luc reporter where the HRE was deleted (HREΔ) or mutated to CAtaTG (m1) or tcacgG (m2) and evaluated their activity in Jurkat cells cultured under 20% or 1% $O_2$ conditions. Mutating or deleting the HRE in the LTR significantly reduced luciferase expression at both oxygen tensions (Fig. 5c), suggesting this motif or sequence regulates HIV transcription most likely by binding of other host transcription factors, possibly CAAAT/EBP US2[38]. Notably, although low oxygen reduced wild-type (WT) LTR activity it had no effect on the LTR HRE mutants (Fig. 5c), suggesting a role for this motif in regulating the promoter activity under a hypoxic environment. Although the mutated LTR sequences showed reduced activity under normoxic conditions

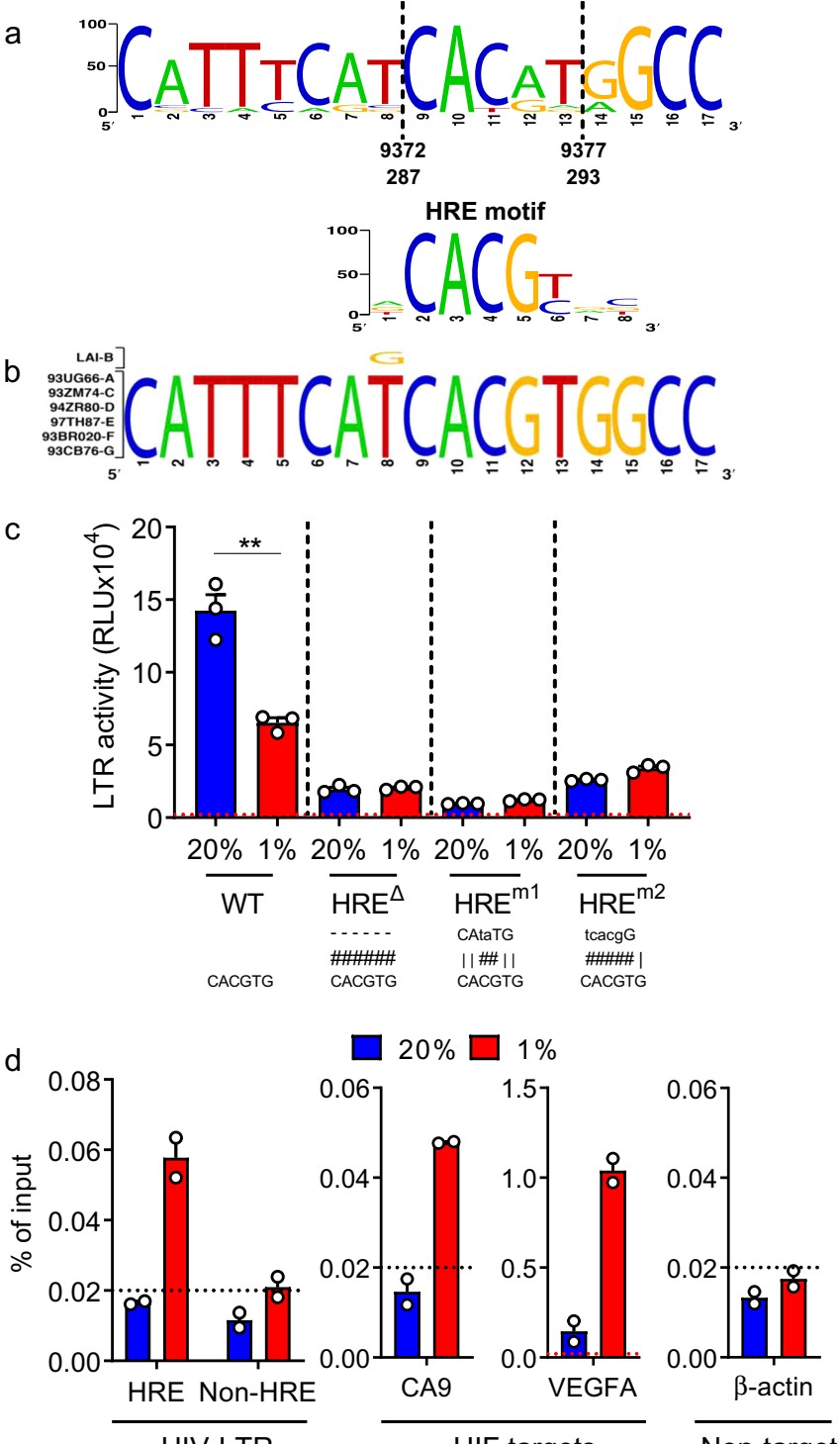

**Fig. 5 Conserved HRE in HIV-LTR. a** Consensus plot illustrating the conserved nature of the HRE in the HIV-1 LTR based on the 897 HIV-1 sequences available in the LANL repository (analyzed with LANL QuikAlign/AnalyzeAlign software, coordinates are from the HXB2 referent). The level of conservation is reflected by the height of the bases (y-axis 0—100%) with the consensus HRE motif (JASPAR database)[90] shown below. The level of conservation is reflected by the height of the bases (y-axis 0—100%)[91,92]. **b** Conserved HRE motifs in the HIV-1 strains studied in Figs. 3–4, where HIV LAI has a variant, G, at position 8. **c** HRE mutation reduces HIV-LTR activity and abrogates the low-oxygen phenotype. Jurkat cells were transfected with WT or HRE mutated HIV-LTR constructs (where hash represents a deletion or mutation) and the cells cultured under 20% or 1% $O_2$ conditions for 48 h. LTR activity was measured and is presented as relative light units (RLU). The dashed line represents the mean RLU of nontransfected cells cultured under 20% and 1% $O_2$ (n = 3, mean ± SEM, Two-way ANOVA analysis). **d** HIF-1β binds the HIV-1 LTR. VSV-G complemented HIV NL4.3 R-E- infected Jurkat cells were cultured at 20% or 1% $O_2$ for 24 h and HIF-1β binding to the HIV-LTR or known HIF target genes carbonic anhydrase IX (CA9) or vascular endothelial growth factor A (VEGFA) was monitored by ChIP. Data are expressed as a percentage of input DNA recovered. Data represent two independent ChIP experiments. Each symbol represents the mean of three technical replicates (n = 2, mean ± SEM, Two-way ANOVA analysis).

there was still a signal (dashed line) above the background to observe any repressive effects of low oxygen. To determine whether HIFs interact with the promoter in HIV-infected Jurkat cells cultured under 20% or 1% $O_2$ we isolated chromatin and showed HIF-1β binding to the LTR under low-oxygen conditions (Fig. 5d). We studied HIF-1β binding since the antibody recognizing this protein has been used extensively for ChIP studies and enables the measurement of total HIF-1β/HIF-α bound complexes. As a specificity control we demonstrated HIF-1β binding to the HIF target genes, carbonic anhydrase IX (CA9) and vascular endothelial growth factor (VEGFA), and negligible binding to β-actin (Fig. 5d). Collectively, these results show that low oxygen inhibits basal and Tat-activated HIV-LTR activity in a HIF-dependent manner.

**Low oxygen inhibits latent HIV reactivation**. The reservoir of latently-infected cells present in ART-treated HIV-1 infected individuals remains a major barrier to cure. To understand the impact of low oxygen on reactivation of latent HIV we employed the well-established Jurkat latency model (J-Lat). The J-Lat cell line contains a full-length integrated HIV-1 genome that encodes GFP, providing a fluorescent marker of HIV-1 transcriptional reactivation (Fig. 6a). Treating cells with TNFα or the clinically approved histone deacetylase (HDAC) inhibitor, Romidepsin, activated latent HIV and induced GFP expression. Under low-oxygen conditions we noted that the frequency of reactivated cells and their levels of GFP expression were significantly reduced (Fig. 6b, c), in the absence of any detectable effect on cellular viability or activation. We confirmed these observations in an independent latently-infected T-cell line (ACH2), showing a significant reduction in p24 antigen and HIV RNA levels following TNFα treatment under low-oxygen conditions (Supplementary Fig. 6). As expected the hypoxic mimetic DMOG significantly reduced the frequency of TNFα or Romidepsin reactivated cells (Fig. 6d). Treating TNFα activated J-Lat cells with the HIF pathway inhibitor NSC-134754 abrogated the inhibitory effect of low oxygen, demonstrating that HIFs limit HIV reactivation under hypoxic conditions (Fig. 6e). In contrast, NSC-134754 reduced Romidepsin-dependent HIV reactivation, that is most likely explained via indirect effects of the HDAC inhibitor on HIF-transcriptional activity[39]. Under low-oxygen conditions NSC-134754 and TNFα treatment induced higher levels of HIV reactivation than Romidepsin (Fig. 6e). These results have important implications for strategies aiming to reactivate latent virus in order to render it accessible for targeting by the immune system and antiretroviral drugs.

## Discussion

In the current study, we show that HIV-1 transcription in activated CD4 T cells is reduced under low-oxygen conditions compared to standard laboratory conditions. Silencing HIF-2α or inhibiting HIF signaling reversed the dampening effect of low oxygen on the viral promoter, suggesting a role for HIF-2α in repressing HIV-1 transcription in CD4 T cells. Furthermore, we show a role for low oxygen in repressing HIV reactivation in latently-infected T cells and in limiting the efficacy of the HDAC inhibitor Romidepsin. We demonstrate that a conserved HRE in the U3 region of the HIV-1 LTR can bind HIF-1β under hypoxic conditions. Our observations are consistent with a previous report[40] showing reduced HIV RNA levels in T cells cultured under 3% $O_2$; however, this study did not investigate a direct role for HIFs in regulating viral replication. The authors concluded that the reduced levels and/or activity of CDK9 and cyclin T1 under low oxygen reduced Tat-mediated transcription (reviewed in ref. [41]). This pathway may work in concert with HIF-2α to

repress HIV-1 replication; however, our observation that 1% oxygen reduces HIV-1 LTR activity independent of Tat (Fig. 3b), suggests that these pathways are independent. We recognize the difficulties in modelling oxygen tension of lymphoid organs that may experience a gradient of oxygen using in vitro models. We selected to use 1% oxygen to ensure HIF stabilization and to be consistent with other published reports studying the role of hypoxia in T-cell function[34,35,42].

HIFs are the major coordinators of the cellular response to hypoxia and are recognized to activate transcriptional responses; however, our current understanding of the mechanisms underlying hypoxia-induced gene repression are limited (reviewed in ref. [43]). HIF binding could displace known transcriptional activators such as the MYC Proto-Oncogene or recruit corepressors such as SIN3A[44]. Recent bioinformatic studies report that SIN3A and a number of its corepressors including HDAC1 are overrepresented in genes transcriptionally repressed by hypoxia[44,45]. Interestingly, silencing HIF isoforms limited the majority of gene induction and repression, suggesting a role for HIFs in transcriptional repression possibly via trans-acting elements[46,47]. Since dioxygenases regulate several aspects of gene expression such as histone methylation and DNA methylation[37], one could hypothesize a role for these enzymes in repressing HIV gene transcription under hypoxic conditions. Of note, we show that DMOG, a competitive antagonist of 2-OG and broad-spectrum inhibitor of the 2-OG oxygenases[48], inhibits HIV-LTR activity and viral reactivation in the J-Lat model, highlighting a potential role for these oxygenases to work in concert with HIFs to regulate HIV replication.

Although HIV exhibits tremendous sequence diversity, the LTRs are relatively well-conserved within and between HIV-1 group M viral subtypes due to their critical biological function (www.hiv.lanl.gov). The HIV-1 5'LTR is ~640 bp in length and the U3 region is located upstream of the transcription start site and is divided into the modulatory (−454, −105), enhancer (−105, −79), and promoter (−78, −1) regions (numbering based on the HXB2 sequence)[49]. The modulator region contains several transcription factor binding sites[50], the enhancer region contains binding sites for NF-kB family members and the core promoter encodes three binding sites for Sp1 and the TATA box. The HIF binding site is located in a region around the −246 position in the U3 region relative to the start codon, and overlaps with a CCAAT/enhancer binding site (C/EBP binding site US3, HXB2 numbering)[38,51]. The HRE is conserved in approximately one third of HIV-1 LTR sequences reported in the LANL database[52]. Of note, 43/84 (~50%) LTR sequences derived from plasma viruses present in individuals in the acute stages of infection (Fiebig stages F1-5[53]) and deduced to reflect transmitted-founder viruses encode a HRE (Supplementary Fig. 7). Transmitter-founder viruses have a unique phenotype that reflects selection for optimal fitness at the initial sites of viral replication[54,55], raising questions about the local oxygen tension and HIF expression levels at these sites. A recent study reported an association between a hypoxic gene signature and inflammasome pathways in CD14+CD16+ monocytes in macaques immunized with candidate HIV vaccine vectors and a decreased risk of SIV$_{mac251}$ acquisition[56], suggesting a role for hypoxia in restricting viral transmission. It is interesting to note that a HRE identified in the LTR of the human endogenous retrovirus HERV-E was reported to be a key modulator of retroviral expression in clear cell Renal Cell Carcinoma[57]. As viral genetic factors determine at least in part the course of disease progression, further studies to investigate the impact of low oxygen in regulating diverse HIV-1 strains are warranted.

HIFs regulate T-cell survival, differentiation, and proliferation under normoxic conditions via T-cell receptor engagement[58–60]

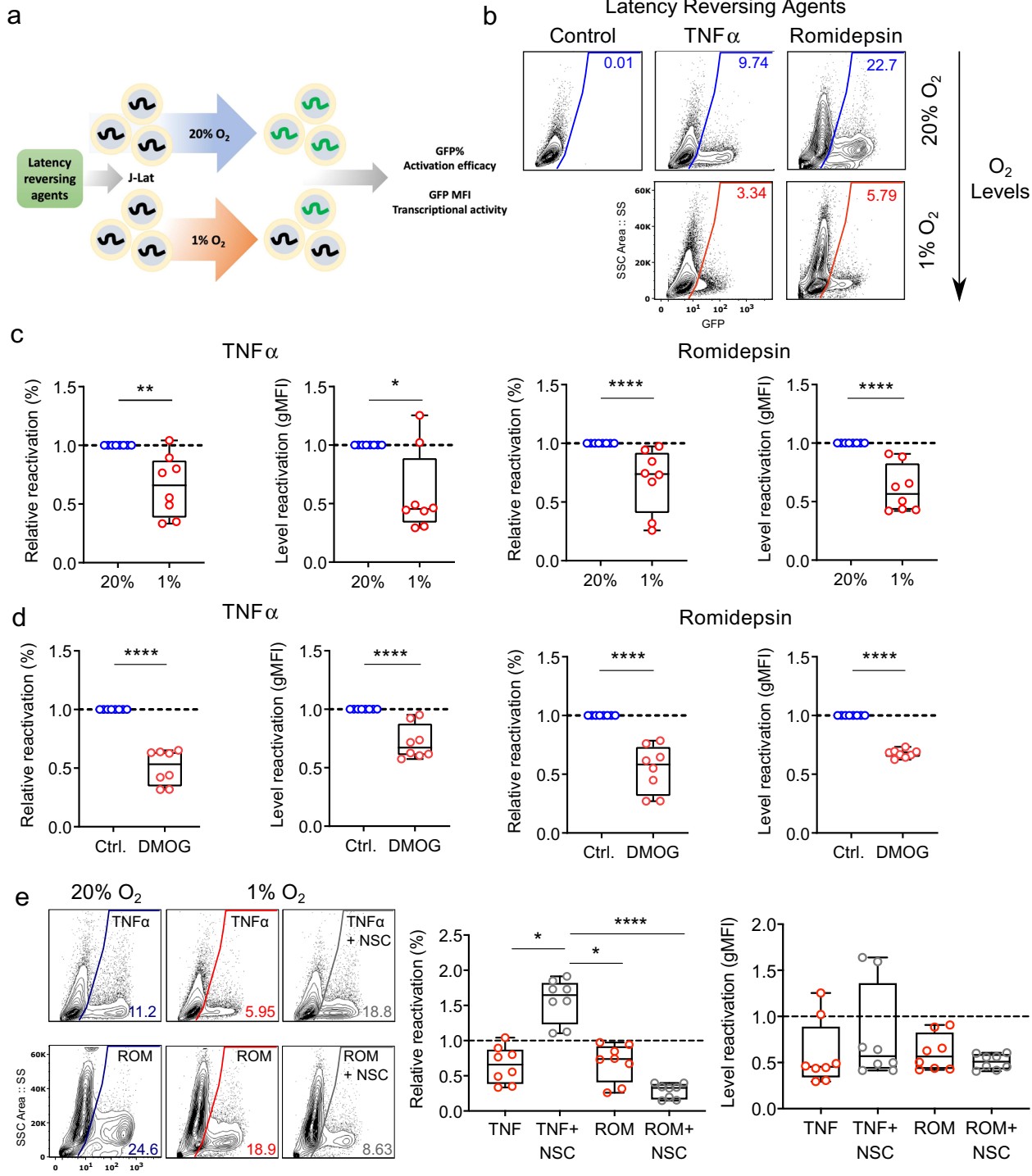

**Fig. 6 Low oxygen reduces HIV reactivation from latency. a** Diagram depicting the protocol used to study the effect of low oxygen on HIV-1 reactivation from latency. In all experiments, the frequency of reactivated cells (% GFP + cells) and their mean GFP expression (gMFI) were assessed by flow cytometry analysis. **b, c** Low oxygen reduces HIV reactivation. J-Lat cells were treated with TNFα or Romidepsin to activate HIV-1 under 20% or 1% $O_2$ conditions. **b** Contour plots with outliers illustrating representative examples of GFP expression in the unstimulated and activated cultures at 20% or 1% $O_2$; the positive gate was defined based on unstimulated control cells. **c** Summarized data (eight replicates from four independent experiments) are shown; results are expressed relative to the control (20% $O_2$) cells and group medians, range and quartiles are indicated (Mann–Whitney analysis). **d** DMOG reduces HIV reactivation. J-Lat cells were treated with TNFα or Romidepsin to activate HIV in the presence or absence of DMOG (0.5 mM) for 24 h. Summarized data ($n = 8$) are shown; results are expressed relative to the control (non-DMOG treated cells) and group medians, range and quartiles are indicated (Mann–Whitney analysis). **e** HIF pathway inhibitor restores HIV reactivation under 1% $O_2$. J-Lat cells were treated with TNFα or Romidepsin to activate HIV under 20% or 1% $O_2$ in the presence or absence of NSC-134754 (1 μM) (NSC). Contour plots with outliers illustrating representative examples of GFP expression in the unstimulated and activated cultures with and without NSC are shown on the left; the positive gate was defined based on unstimulated control cells. Summarized data ($n = 8$) from both untreated and NSC-treated cells cultured under 1% $O_2$ are expressed relative to cells cultured under 20% $O_2$ and group medians, range and quartiles are shown on the right (Mann–Whitney analysis).

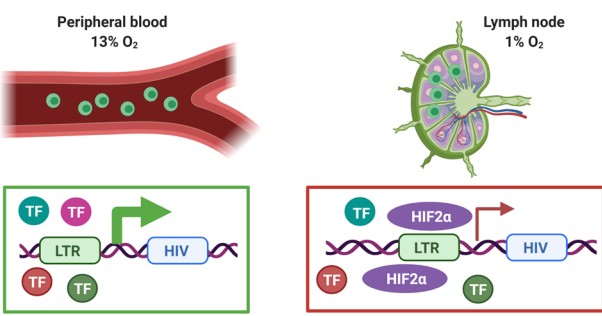

**Fig 7** Model of how the hypoxic environment of the lymph node may suppress HIV replication and promote latency.

and generation of reactive oxygen species (ROS)[14,61]. HIF-1α induces a shift in cellular metabolism from oxidative phosphorylation toward glycolytic pathways, enabling effector function to be sustained with minimal oxygen consumption[62]. Chronic immune activation and inflammation are hallmarks of HIV-1 infection and Duette reported increased HIF-1α expression in HIV-1 infected T cells via the induction of mitochondrial ROS[63]. The authors evaluated the functional significance of HIF-1α in HIV replication by treating cells with cobalt chloride, a hypoxia mimetic that induces HIF-1α activity, concluding that HIF-1α activated HIV transcription. However, the action of chemical mimetics such as cobalt chloride or flavonoids may be mediated by signaling pathways not necessarily shared by the "true" hypoxic response and may induce different and oxygen-independent biological effects[64,65]. It is noteworthy that Befani et al. reported that cobalt stimulates HIF-1α dependent but inhibits HIF-2α dependent gene expression in liver cells[66], providing a potential explanation for the differing results with our study and highlighting the importance of studying HIV replication under low-oxygen conditions.

Cellular metabolism is known to influence HIV replication: viral encoded accessory proteins increase glucose uptake and expression of glycolytic enzymes in infected cells[67], and partial inhibition of glycolysis impairs virus replication[68]. How the HIF-induced shift in cellular metabolic pathways impacts HIV replication in CD4 T cells remains to be determined: although HIF-1α enhances the survival of activated T cells[69] and upregulates glycolytic pathways[70] that would theoretically support viral replication, the shut-off of oxidative phosphorylation (on which T-cell proliferation is reliant[71]) may create an environment that is not optimal for virus replication. The current study focuses on direct effects of hypoxia on HIV replication, which was evaluated using a system where T cells were activated in normoxic conditions prior to infection and viral replication assessed over a relatively short time-frame. It will be of interest to explore the effects of T-cell activation under conditions of both physiological and pathological hypoxia, the more severe hypoxia induced by conditions including infection and inflammation[72,73] on the replication of diverse HIV-1 strains, comparing those with or without HRE in their LTR promoter regions, in future studies.

The low-oxygen tension in lymphoid tissues is also likely to influence HIV latency and persistence. The pool of latently-infected cells is established early after HIV-1 infection and constitutes a major barrier to HIV-1 eradication and cure[74,75]. Our data support a model where the hypoxic environment of the lymph node may suppress HIV replication and promote latency (Fig. 7). Current strategies to eradicate HIV-1 from individuals on ART are designed to reactivate latent reservoirs and simultaneously employ immune interventions to target and destroy antigen-expressing cells[76,77]. However, such "kick and kill"

approaches have been hampered by the relatively poor in vivo efficacy of the latency-reversing regimes tested to date[26,78,79]. The recent discovery and excellent safety profile of a HIF-2α antagonist in patients with clear cell renal cell carcinoma[80,81] may provide a novel strategy to promote HIV-1 reactivation and combined treatment with latency-reversing agents may provide exciting new therapies.

## Methods

**Cell lines and reagents.** 1G5 cells were provided by Professor Ariberto Fassati (University College London, London, UK) and cultured in RPMI supplemented with 10% heat inactivated (hi)FBS and 1% penicillin/streptomycin. TZM-bl and ACH2 cells were provided by Professor Bill Paxton (University of Liverpool, Liverpool, UK) and cultured in DMEM medium supplemented with 10% hiFBS, 1% penicillin/streptomycin and 1% L-glutamine. Jurkat cells and J-Lat clone 6.3 were provided by Professor Xiaoning Xu (Imperial College, London) and maintained in RPMI media containing 10% hiFBS. 293 T cells were cultured in RPMI media supplemented with 10% hiFBS and 1% penicillin/streptomycin (obtained from ATCC). Antibodies specific for HIF-1α were purchased from BD Biosciences and anti-HIF-2α antibody provided by the Ratcliffe laboratory (University of Oxford). NSC-134754 was previously reported[82,83]. DMOG and Romidepsin were purchased from Cambridge Biosciences, UK. siRNAs specific for HIF-1α (s6539) and HIF-2α (s4700) were purchased from ThermoFisher Scientific, UK.

**Plasmids.** Plasmids encoding full-length infectious molecular clones of NL4.3 and NL4.3-Bal were obtained from Drs John Kappes and Christina Ochsenbauer-Jambor (University of Alabama at Birmingham, USA). HIV NL4.3 R-E- Luciferase was obtained from the NIBSC AIDS Repository. VSV-G expression plasmid was previously reported[84]. HIV Tat expression plasmid and HIV-LTR-Luc constructs were previously reported[85] and provided by Professor Bill Paxton (University of Liverpool, Liverpool, UK). HIV-LTR NF-kB deletion mutant was published[86] and provided by Professor Bassel E Sawaya (Temple University Medical School, US). Mutagenesis of the HIV-LTR was performed using an Agilent mutagenesis kit (Agilent, UK) following the manufacturer's instructions. The HRE luciferase reporter plasmid was provided by the Ratcliffe Laboratory (University of Oxford). pcDNA3.1-EGFP was used to monitor transfection efficiency.

**Primers.** Oligonucleotide sequences are listed in Supplementary Table 1 and were purchased from Life Technologies.

**Generation and titration of viral stocks.** To produce HIV NL4.3 or NL4.3-Bal stocks, plasmid DNA was transfected into 293FT cells using Lipofectamine (Life Technologies, UK) or Fugene 6 (Promega, UK) and virus containing supernatants harvested 3 days later, clarified by centrifugation at $1400 \times g$ for 10 min and stored at $-80\,°C$. The infectivity of viral stocks was determined using a colorimetric reverse transcriptase assay (Roche Life Sciences). VSV-G complemented NL4.3 R-E- viral stocks were generated as previously reported[84] and RT activity measured using a PCR based method[87].

**Viral replication assay.** PBMCs were isolated from leukapheresis samples obtained with appropriate ethical approval from healthy donors who provided written informed consent (NHS Blood and Transplant Service). All work was compliant with institutional guidelines. PBMCs were depleted of CD8+ cells using CD8 microbeads (Miltenyi Biotec) and the cells stimulated with 50 IU/ml IL-2 (Proleukin; Novartis), 0.01μg/ml soluble human anti-CD3 (R&D; clone UCHT1) and 0.1 μg/ml soluble human anti-CD28 (Life Technologies; clone CD28.2) at a concentration of $1 \times 10^6$ cells/ml in complete RPMI-1640 (Life Technologies) containing 10% FBS (Sigma), 1% penicillin-streptomycin (Sigma), 1% sodium pyruvate (Sigma), 1% Glutamax (Life Technologies), 1% nonessential amino acids (Life Technologies), and 2 mM beta-mercaptoethanol (Life Technologies), and incubated at 37 °C in 5% $CO_2$ for 3 days. After initial activation, $4–5 \times 10^6$ cells were infected with HIV NL4.3-Bal (2.5 ng/ml RT activity per $1 \times 10^6$ cells) by spinoculation (2 h, $1200 \times g$, 20 °C) in a final volume of 100 μL. Cells were washed twice and resuspended in complete RPMI + 50 IU/ml IL-2 at $1 \times 10^6$ cells/ml and plated in U-bottomed 96-well plates (Corning) in a final volume of 200 μL/well. All conditions were seeded in duplicate. Cells were incubated at either 20% $O_2$ or 1% $O_2$ for 4 days or 1% $O_2$ for 2 days and reoxygenated in 20% $O_2$. At days 2 and 4, extracellular media was collected and p24 antigen levels quantitated using a HIV p24 high sensitivity AlphaLISA kit (Perkin Elmer, UK).

**Flow cytometry.** To measure Glut-1 expression, $2 \times 10^5$ uninfected preactivated cells were collected after 2 or 4 days of exposure to 20% or 1% $O_2$ levels, incubated with Human TruStain FcX (FC receptor blocking solution; BioLegend) and stained with an antibody cocktail: anti-CD4 APC-H7 (SK3, BD Biosciences, 1:50 dilution); anti-CD38 Pe-Cy5 (HIT2, BioLegend, 1:500 dilution); anti-HLA-DR BV421 (L243, BioLegend, 1:200 dilution) and Live Dead Aqua to measure cell viability (Life Technologies, 1:1000 dilution). Cells were fixed with BD Cytofix/Cytoperm buffer

(BD Biosciences), washed and resuspended in 200 μL PBS overnight. The following day, cells were permeabilized and stained with anti-CD3 PE-Texas Red (S4.1, Life Technologies, 1:50 dilution) and anti-GLUT-1 APC (202915, R&D, 1:25 dilution). Cell fluorescence was detected on a BD Fortessa X20 flow cytometer using BD FACSDiva8.0 (BD Bioscience) and data analyzed using FlowJo 10 (BD Biosciences). For in-depth activation and viral entry receptor analysis, $2 \times 10^5$ pre-activated cells were collected after 2 or 4 days of exposure to 20% or 1% $O_2$ levels, incubated with Human TruStain FcX (FC receptor blocking solution; BioLegend, 1:50 dilution) and stained with an antibody cocktail: CCR5 Pe-Dazzle 594 (J418F1, BioLegend, 1:200 dilution), CCR6 PE (11A9, BD Biosciences, 1:50 dilution), CCR7 BV650 (G043H7, BioLegend, 1:100 dilution), CD134 BB700 (ACT35, BD Biosciences, 1:100 dilution), CD25 PE-Cy7 (2A3, BD Biosciences, 1:100 dilution), CD38 AF488 (HIT2, BioLegend, 1:200 dilution), CD4 APC-H7(SK3, BD Biosciences, 1:50 dilution), CD45RA BV711 (HI100, BioLegend, 1:200 dilution), CXCR3 PE-Cy5 (1C6/CXCR3, BD Biosciences, 1:50 dilution), CXCR4 BV605 (12G5, BioLegend, 1:200 dilution), HLA-DR BV421 (L243, BD Biosciences, 1:200 dilution), PD-1 BV786 (EH12.1, BD Biosciences, 1:200 dilution), and Live Dead Aqua to measure cell viability (1:1000 dilution). Cells were fixed and permeabilized with Foxp3 Fixation/Permeabilization kit (Life Technologies), stained with CD3 APC-R700 (UCHT1, BD Biosciences, 1:50 dilution) and Foxp3 APC (PCH101, Life Technologies, 1:50 dilution) for 45 min. Cells were washed and resuspended in 200 μL PBS. Cell fluorescence was detected on a BD Fortessa X20 flow cytometer using BD FACSDiva8.0 (BD Bioscience) and data analyzed using FlowJo 10 (BD Biosciences).

**Quantification of HIV-1 reverse transcription products and integration**. Total DNA was extracted from HIV-infected cells using a DNeasy kit (Qiagen) and yields quantified by UV spectroscopy (Nanodrop 100, ThermoFisher). To monitor the infection process we performed a quantitative PCR using primers specific for the earliest 'Strong Stop' products of HIV reverse transcription using a minor modification of previously published methods[88]. A 134 bp region within the early transcript between 499 and 633 of the HIV-LTR (HXB2 numbering, Los Alamos) was amplified using two primers and detected with a TaqMan probe which binds to the region between 554 and 576 (HXB2 numbering, Los Alamos). PCR reactions were set up using 500 ng of genomic DNA in accordance with the manufacturer's instructions (MyTaq, Bioline). PCR amplification of the 1st (minus) strand transfer and the 2nd (plus) strand transfer cDNA synthesis products were performed using primers oHC65 and HIV-fst-f1 (amplifies a 226 bp region between nucleotides 409 and 635) and oHC64 and gag-m661-as (amplifies a 195 bp region between nucleotides 499 and 694), respectively. The TaqMan probe oHC66 was included in both qPCRs, with the same reaction conditions as described above.

HIV-1 integration in the infected cultures was assessed using a minor modification of previously published methods[89]. Briefly, a nested PCR was performed using an initial PCR with primers specific for Alu repeat regions of the human genome and for the region 1502-1486 of the HIV gag gene. These PCR reactions were set up using 500 ng of genomic DNA in accordance with the manufacturer's instructions (MyTaq, Bioline). A second PCR was set up using 3 μL of the above completed PCR as template and with primers targeting a region of the HIV-1 LTR present within the first amplicon. The primers were located within the LTR at 521-539 and at 644-628. The TaqMan probe oHC66 was included in the qPCR, with the same reaction conditions as described above. In these assays a calibration standard of genomic DNA prepared from ACH2 cells was used.

**HIV promoter activity**. 1G5 and TZM-bl LTR-Luc reporter cells were spinocu-lated with HIV-1 NL4.3-Bal or NL4.3 (2 h, $1200 \times g$, 20 °C) using 1 ng/ml of HIV RT activity per $10^6$ cells in a final volume of 150 μL. Cells were washed twice and resuspended in complete RPMI at $1 \times 10^6$ cells/ml in U-bottomed 96-well plates or complete DMEM at $8 \times 10^4$/ml in flat-bottomed 96-well plates, respectively. All conditions were seeded in six biological replicates. After 48 h cells were washed, lysed and luciferase activity measured. To assess the effect of low oxygen on basal and TAT-mediated LTR activity, $2 \times 10^5$ TZM-bl cells were transfected with 2 μg of pTAT plasmid using Fugene transfection reagent (Promega). Twenty-four hour later, the transfected cells were cultured under 20% or 1% $O_2$ conditions for 48 h and luciferase activity measured. HIV-1 LTR-Luc plasmids (2 μg) with or without siRNA (50 nM) were delivered into Jurkat cells using the Neon® Transfection System following the manufacturer's instructions. To monitor transfection efficiency the HIV-LTR plasmids were co-transfected with pcDNA3.1-EGFP (0.1 μg) and samples analyzed by a Cyan ADP flow cytometer (Beckman Coulter) and data analyzed using FlowJo 9.9.4 (TreeStar). Transfected cells were lysed in RIPA buffer (20 mM Tris, pH 7.5, 2 mM EDTA, 150 mM NaCl, 1% NP-40, and 1% sodium deoxycholate) supplemented with protease inhibitor cocktail tablets (Roche). 4× reducing buffer was added to samples before incubating at 95 °C for 5 min. Proteins were separated on a 10% polyacrylamide gel and transferred to PVDF membranes (Amersham). The membranes were blocked in PBST, 5% skimmed milk (Sigma), and proteins detected using HIF-1α or HIF-2α antibodies at 1 in 1000 dilution and HRP-secondary antibodies. Protein bands were detected using Pierce SuperSignal West Pico chemiluminescent substrate kit (Pierce) and images collected with a PXi Touch gel and membrane Imaging system (Syngene).

**HIV-LTR HRE analysis**. The HIV-1 LTR sequences deposited in the Los Alamos Database were searched for the HRE motif RCGTGC using the Sequence Search Interface program (www.hiv.lanl.gov) and the number and location of matches enumerated.

**ChIP and quantitative PCR**. Jurkat cells were infected with VSV-G-pseudotyped HIV-1 NL4.3 R⁻E⁻ and 3 days post-infection the cells cultured at 20% or 1% $O_2$ at $1 \times 10^6$ cells/ml for 24 h. Cells were fixed with 1% formaldehyde (Sigma Aldrich 47608) for 10 min at room temperature before quenching with 125 mM glycine. Cells were washed with ice cold PBS containing EDTA-free protease inhibitors (Roche)/5 mM Na-butyrate and lysed for 1 min at 4 °C in 500 μL of Nuclear Extraction buffer (10 mM Tris-HCl (pH 8.0), 10 mM NaCl, 1% NP-40) supplemented with protease inhibitor cocktail (Roche). The nuclei were recovered by centrifugation at $500 \times g$ for 5 min at 4 °C and resuspended in 100 μl lysis buffer (50 mM HEPES/KOH, pH 7.5, 140 mM NaCl, 1 mM EDTA, 1% Triton X-100) and pulse sonicated using a Bioruptor® sonicator (Diagenode, UK) at high power for 30 min on ice (15 sec on, 15 sec off). The sonicated lysates were clarified by centrifugation at $16,000 \times g$ for 10 min and HIF-1β complexes immunoprecipitated as per manufacturer's instructions using a ChIP-IT® Express Chromatin Immuno-precipitation kit, including Protein G Magnetic Beads (Active Motif, USA). The input and recovered DNA were quantified by real-time PCR using a Lightcycler 96 (Roche) PCR System. The values were calculated as the ratio between the ChIP signals and their respective input DNA signals and expressed relative to their normoxic samples determined by the $2^{-\Delta\Delta Ct}$ method.

**HIV latency assays**. J-lat or ACH2 cells were resuspended in complete RPMI at a concentration of $2 \times 10^6$ cells/ml and stimulated with TNFα (100 μg/ml) or Romidepsin (20 nM) in the presence or absence of DMOG (0.5 mM) or NCS-134754 (1 μM). Cells were plated into U-bottomed 96-well plates (Corning) in a final volume of 200 μL/well and incubated at 20% $O_2$ or 1% $O_2$. After 24 h cells were stained for 10 min at room temperature with Live Dead Aqua to measure cell viability (Life Technologies) and fixed with 4% PFA (Santa Cruz) for 10 min at room temperature. All samples were acquired on a Cyan ADP flow cytometer (Beckman Coulter) and data analyzed using FlowJo 9.9.4 (TreeStar).

**Statistics and reproducibility**. All experiments were performed at least three times. All data are presented as mean values ± SEM. $P$-values were determined using the Mann–Whitney test (two group comparisons; unpaired data) or with the Wilcoxon matched-pairs signed rank test (two group comparison; paired data) using PRISM version 8. In the figures single asterisk denotes $p < 0.05$, double asterisks denote $p < 0.01$, triple asterisks denote $p < 0.001$, quadruple asterisks denote $p < 0.0001$, n.s. denotes non significant.

**Reporting summary**. Further information on research design is available in the Nature Research Reporting Summary linked to this article.

## Data availability
The authors declare that all data supporting the findings of this study are available in the article along with supplementary Information file. Source data are available in Supplementary Data 1.

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

## Acknowledgements

We thank our colleagues Andrew McMichael; William James, Lucy Dorrell, Jan Rehwinkel and Alvina Lai for critical reading of the manuscript and Alvina Lai for assistance with drawing figures. We thank John Kappes and Christina Ochsenbauer-Jambor for the HIV-1 NL4.3 and NL4.3-Bal infectious molecular clones; Bill Paxton (University of Liverpool) for diverse HIV-LTR-Luc plasmids, TZM-bl and ACH2 cells; Ariberto Fassati (UCL, London) for 1G5 cells; Xiaoning Xu (Imperial College, London) for Jurkat and J-Lat clone 6.3 cells and Bassel Saway (Temple University Medical School, US) for the NF-Kb deleted HIV-1 LTR-reporter plasmid.

## Author contributions

X.Z. designed and conducted experiments and co-wrote the MS; I.P-P. conducted experiments and edited MS; I.N. conducted experiments; A.E.K. conducted experiments; A.M. advised on LTR mutagenesis protocols; W.P. advised on plasmid delivery protocols; C.O.R. generated cell and virus stocks; H.Y. advised on latency model; M.A. provided reagents; D.M. provided reagents and advice and edited the MS; P. Balfe designed experiments, conducted analysis and co-wrote MS; P.B. designed experiments and co-wrote MS; J.A.M. designed study and co-wrote MS.

## Competing interests

The authors declare no competing interests.
