## [Peer Review File · Communications Biology]

Reviewers' comments:

Reviewer #1 (Remarks to the Author):

Summary:

The authors describe the impact of hypoxia on viral replication through hypoxia inducible factor signaling (HIF). HIF bind to hypoxia response elements (HRE) in the promoter/enhancer elements of target genes. They found that hypoxia results in repressed HIV replication and a reduction in reactivation of latent HIV. The HIF induced gene GLUT-1 was used as confirmation as to whether the HIF pathway was activated. The low oxygen state affected post-integration steps only. The study also accounted for the fact that CD4 T cells migrate in and out of lymphoid tissues therefore being exposed to both oxygenated blood as well as lower oxygen environments of lymphoid tissue. The effect occurred with both X4 and R5 tropic viruses and multiple clades. Gene silencing demonstrated HIF-1a as the primary factor involved. A HRE was found in the HIV LTR. This is a very well-designed study with convincing results. This study has important implications for improving the efficacy of latency reversal agents.

Major Comments:

1. The authors should demonstrate that first oxygenated and then underoxygenated cells will have the same effect of repressing HIV replication as they have shown the reverse direction of underoxygenation followed by oxygenated conditions.
2. Although the relative changes are compelling, it would be prudent to demonstrate the absolute replication quantity for the baseline models to ensure they are adequate and comparable to other similar systems. This would also help to establish whether changes with the knock down model and inhibitor models appear similar to the baseline models.
3. The authors should comment on the impact of low oxygen on CDK9 activity, p27 expression and PP1 activity that are either separate or related to the involvement of HIF.

Minor Comments:

1. The authors should state whether they explored slightly higher oxygen conditions (2-4%) to see if there is a threshold or dose dependent effect.
2. The use of either "a" or "α" for "alpha" from HIF1a and HIF2a should be standardized.

Reviewer #3 (Remarks to the Author):

Brief summary:

In the submitted manuscript, Zhuang et al demonstrate through a series of in vivo experiments that low oxygen tension reduces HIV replication, and that this is mediated through HIF-dependent mechanisms. These in vitro studies provided good evidence that HIF-2, consisting of the HIF-2a/HIF-1b dimer, plays an important role in decreasing HIV transcription through the direct interaction of HIF-2a with the HIV LTR through a conserved HRE buried in the U3 LTR.

Overall impression of the work:

Zhuang et al have carried out an elegantly designed series of studies that has the potential to be a game-changer for the HIV cure field. Although other groups have shown decreased HIV transcription in low oxygen conditions (Charles et al, J Cell Physiol 2009 Nov;221(2):469-79), this work goes a few steps farther by investigating the roles of the various hypoxia inducible factors in the context of HIV infection. Crucially, this paper also drives home the importance of hypoxia-induced changes in HIV transcription in lymphoid tissues, which at this point are well established as an important microanatomic reservoir for HIV latency. This could have real impact when designing future translational studies aimed at reactivating latent virus in lymphoid tissues, since this reservoir has been extremely difficult to target. The manuscript was very well written and well organized, and in my opinion deserves to be published with minimal changes.

Specific comments:

1. I would find it helpful, as a non hypoxia expert, for the authors to expand a little upon the currently known roles of the various HIFs in HIV replication. For instance, there is at least one recent manuscript that links HIF-1 α with increased viral replication, which, at least generally, appears to be at odds with the results of this study (see Duette et al Induction of HIF-1 α by HIV-1 Infection in CD4+ T Cells Promotes Viral Replication and Drives Extracellular Vesicle-Mediated Inflammation. MBio. 2018 Sep 11;9(5). pii: e00757-18). Can the authors comment on some of the discrepancies in the literature?
2. On page 6, lines 12-13 there is a sentence fragment, "We selected to study HIF-1b binding since the antibody recognizing this protein has been used extensively for ChIP studies..." I'm there should be a word after "selected" but I'm not sure what the authors were referring to.
3. The authors use both "a" and " α " in the text when referring to HIF-1a and HIF-1b. Which is most correct?

Reviewer #4 (Remarks to the Author):

Most cells in our body are maintained in ~3–6% oxygen. Yet, most studies of human immunodeficiency virus (HIV) replication and latency are performed in vitro under ~20% oxygen, non-physiological conditions, where hypoxia inducible factors (HIFs) are inactive. The role of physiological factors (such as hypoxia) in the regulation of HIV transcription is not well known. Zhuang et al. investigated the effect of physiological oxygen tension and HIF signaling on HIV-1 replication in CD4 T cells and latency models. The authors demonstrated that hypoxic conditions suppress HIV replication. These findings were already reported by Charels et al., (2009) who showed that HIV-1 transcription in T cells is reduced at 3% oxygen compared with 21% concentration. However, the authors in the current study investigated the mechanism underlying the effect of hypoxia and role of HIFs on HIV-1 replication and showed that HIF-2 restrict HIV transcription via direct binding to its viral promoter and demonstrated that hypoxia reduce the reactivation of latent HIV. While interesting this the following points need to be addressed to make this study more compelling.

Major points:

Figure 1: The authors demonstrate that HIV-1 infection of peripheral blood mononuclear cells (PBMCs) that were cultured under low oxygen, reduces HIV replication. In Supp figure 2, the authors showed also that lower oxygen conditions had an effect on T cell activation status. How can authors rule out that the reduction in HIV replication observed under hypoxia is not due to reduced T cell activation?

Figure 2: Data shown suggests that low oxygen environment is unlikely to perturb HIV-1 entry to CD4 T cells by assessing the expression of the cell-surface receptors that are required for viral binding and internalization (i.e. CD4, CCR5 and CXCR4). However, T cells were cultured at 1% O₂ for only 24h while all experiment performed in Figure 1 were carried out (at 1% O₂) for 2 and 4 days. The authors should monitor the expression of these receptors after 2 and 4 days of hypoxia. In panel C, the authors concluded that the frequency of HIV integration events was comparable under 20% or 1% O₂ conditions. Yet, it is difficult to make such conclusion from only two individual donors. A statistically significant number of donors should be used (HIV integration events seems to be increased under 1% O₂).

Figure 3: The authors evaluated the effect of 1% O₂ on HIV promoter activity using Jurkat 1G5 and HeLa TZM-bl cell lines -commonly used lines to monitor HIV-1 replication. Could the authors confirm that low oxygen conditions have no effect on the early steps of viral life cycle and HIV integrations events in these cells?

Figure 4: Authors show that inhibition of HIF signaling restored HIV LTR promoter activity under low oxygen conditions. Could the authors confirm that the knock-down of HIF-1 α or HIF-2 α or treatment with NSC and 2-OG have no effect on cell viability? Given that HIFs also play a fundamental role in coordinating cellular metabolism and function in T cells, such changes must be excluded. Also, these results should be confirmed in primary CD4 T cells. In panel b, the reduction in HIF-2 α expression upon NSC treatment shown by western blot is not clear.

Figure 5: To determine whether HIFs interact with the promoter in HIF infected Jurkat cells under 1% O₂, the authors performed ChIP experiment using HIF-1 α antibody. However, it is unclear why the authors used HIF-1 α antibody and not HIF-2 α since they showed in figure 4 that only HIF-2 α (and not HIF-1 α) knock-down was able to reverse the phenotype?

Figure 6: The authors demonstrate that low oxygen conditions inhibit latent HIV reactivation using the established Jurkat latency model. It is unclear why the mean GFP expression (gMFI) is not statistically significant. Also, where is the data showing that "under low oxygen conditions NSC-134754 and TNF α induced higher levels of HIV reactivation than Romidepsin" as the authors stated?

Minor points:

- Typo "HIF-2 α " line 4 in the discussion.

Reviewer #1: The authors describe the impact of hypoxia on viral replication through hypoxia inducible factor signaling (HIF). HIF bind to hypoxia response elements (HRE) in the promoter/enhancer elements of target genes. They found that hypoxia results in repressed HIV replication and a reduction in reactivation of latent HIV. The HIF induced gene GLUT-1 was used as confirmation as to whether the HIF pathway was activated. The low oxygen state affected post-integration steps only. The study also accounted for the fact that CD4 T cells migrate in and out of lymphoid tissues therefore being exposed to both oxygenated blood as well as lower oxygen environments of lymphoid tissue. The effect occurred with both X4 and R5 tropic viruses and multiple clades. Gene silencing demonstrated HIF-1a as the primary factor involved. A HRE was found in the HIV LTR. This is a very well-designed study with convincing results. This study has important implications for improving the efficacy of latency reversal agents.

Major Comments:

1.The authors should demonstrate that first oxygenated and then underoxygenated cells will have the same effect of repressing HIV replication as they have shown the reverse direction of underoxygenation followed by oxygenated conditions. Response: Our major goal for these experiments was to study HIV-1 replication in a low oxygen environment eg lymph node and to compare to normal laboratory conditions of 20% oxygen. Having demonstrated that low oxygen reduced HIV-1 replication we designed experiments to assess whether this viral phenotype was reversed following re-oxygenation, to model an in vivo situation where infected CD4 T cells may migrate from lymphoid tissues to a higher oxygen environment eg blood. We have edited the text (page 3, lines 16-17 and 29) to justify our experimental design and clarified Fig.1a cartoon. Given that in vivo the majority of CD4 T cells reside and become infected within lymphoid tissues, we believe this experimental design provides a more appropriate model than a “reverse” model where cells are infected under highly oxygenated conditions and then transferred to a low oxygen environment.

2.Although the relative changes are compelling, it would be prudent to demonstrate the absolute replication quantity for the baseline models to ensure they are adequate and comparable to other similar systems. This would also help to establish whether changes with the knock down model and inhibitor models appear similar to the baseline models. Response: We have included absolute p24 antigen values for viral replication assays in a new supplementary Fig.1a. The viral replication parameters are comparable to previous reports from our lab (Fenton-May, Retrovirology, 2013) and others (Prince, PNAS 2012; Parrish, PNAS 2013).

3.The authors should comment on the impact of low oxygen on CDK9 activity, p27 expression and PP1 activity that are either separate or related to the involvement of HIF. Response: We have discussed and cited the role of CDK9 and PP1 and potential relevance to our data (Page 6, lines 27-3).

Minor Comments:

1. The authors should state whether they explored slightly higher oxygen conditions (2-4%) to see if there is a threshold or dose dependent effect. Response: Since our study is primarily focusing on the role of HIFs in regulating the HIV promoter, we selected to perform our experiments at 1% oxygen as this concentration is conventionally used to model hypoxia ex vivo and allows comparison to published studies. We have edited the text to state our rationale for selecting 1% oxygen (page 3, lines 16-17). The justification for studying different oxygen conditions relates to the requirement of the HIF PHD for molecular oxygen (Kd) and these enzymes can be inactivated at oxygen levels < 5% (Sanchez-Fernandez, Biochem J 2013; Hancock, ACS Chemical Bio 2017).

2. The use of either “a” or “α” for “alpha” from HIF1a and HIF2a should be standardized. Response: Edited throughout revised MS text.

Reviewer #3: In the submitted manuscript, Zhuang et al demonstrate through a series of in vivo experiments that low oxygen tension reduces HIV replication, and that this is mediated through HIF-dependent mechanisms. These in

in vitro studies provided good evidence that HIF-2, consisting of the HIF-2 α /HIF-1 β dimer, plays an important role in decreasing HIV transcription through the direct interaction of HIF-2 α with the HIV LTR through a conserved HRE buried in the U3 LTR.

Overall impression of the work: Zhuang et al have carried out an elegantly designed series of studies that has the potential to be a game-changer for the HIV cure field. Although other groups have shown decreased HIV transcription in low oxygen conditions (Charles et al, J Cell Physiol 2009 Nov;221(2):469-79), this work goes a few steps farther by investigating the roles of the various hypoxia inducible factors in the context of HIV infection. Crucially, this paper also drives home the importance of hypoxia-induced changes in HIV transcription in lymphoid tissues, which at this point are well established as an important microanatomic reservoir for HIV latency. This could have real impact when designing future translational studies aimed at reactivating latent virus in lymphoid tissues, since this reservoir has been extremely difficult to target. The manuscript was very well written and well organized, and in my opinion deserves to be published with minimal changes.

Specific comments:

1. I would find it helpful, as a non hypoxia expert, for the authors to expand a little upon the currently known roles of the various HIFs in HIV replication. For instance, there is at least one recent manuscript that links HIF-1 α with increased viral replication, which, at least generally, appears to be at odds with the results of this study (see Duette et al Induction of HIF-1 α by HIV-1 Infection in CD4⁺ T Cells Promotes Viral Replication and Drives Extracellular Vesicle-Mediated Inflammation. MBio. 2018 Sep 11;9(5). pii: e00757-18). Can the authors comment on some of the discrepancies in the literature? Response: We have edited our manuscript (page 7, lines 29-37) to discuss the observations reported by Duette et al and to highlight differences between the studies.

2. On page 6, lines 12-13 there is a sentence fragment, “We selected to study HIF-1 β binding since the antibody recognizing this protein has been used extensively for ChIP studies...” I’m there should be a word after “selected” but I’m not sure what the authors were referring to. Response: We changed to ‘We studied HIF-1 β binding...’

3. The authors use both “a” and “ α ” in the text when referring to HIF-1 α and HIF-1 β . Which is most correct? Response: We have revised text to ensure a standardized nomenclature.

Reviewer #4: Most cells in our body are maintained in ~3–6% oxygen. Yet, most studies of human immunodeficiency virus (HIV) replication and latency are performed in vitro under ~20% oxygen, non-physiological conditions, where hypoxia inducible factors (HIFs) are inactive. The role of physiological factors (such as hypoxia) in the regulation of HIV transcription is not well known. Zhuang et al. investigated the effect of physiological oxygen tension and HIF signaling on HIV-1 replication in CD4 T cells and latency models. The authors demonstrated that hypoxic conditions suppress HIV replication. These findings were already reported by Charles et al., (2009) who showed that HIV-1 transcription in T cells is reduced at 3% oxygen compared with 21% concentration. However, the authors in the current study investigated the mechanism underlying the effect of hypoxia and role of HIFs on HIV-1 replication and showed that HIF-2 α restrict HIV transcription via direct binding to its viral promoter and demonstrated that hypoxia reduce the reactivation of latent HIV. While interesting this the following points need to be addressed to make this study more compelling.

Major points:

Figure 1: The authors demonstrate that HIV-1 infection of peripheral blood mononuclear cells (PBMCs) that were cultured under low oxygen, reduces HIV replication. In Supp figure 2, the authors showed also that lower oxygen conditions had an effect on T cell activation status. How can authors rule out that the reduction in HIV replication observed under hypoxia is not due to reduced T cell activation? Response: We agree with the reviewer that low oxygen may affect T cell activation status that could impact HIV replication. Indeed, our in vitro system stimulated CD4 T cells for three days before infecting with HIV and assessing the impact of low oxygen on viral replication. Our

initial MS submission focused on the mid-late activation marker CD38. To extend this analysis we designed a 14-colour multiparameter flow cytometry panel to quantify expression of a range of activation markers on CD4 T cells. Our new data from ten different donors after 2 and 4 days of culture under 20% or 1% O₂ conditions show that expression of several CD4 T cell activation markers including CD25, PD-1 and to a lesser extent CD38 are increased following culture in 1% O₂, although other markers such as HLA-DR and CD134 remain unchanged. These data show that CD4 T cell activation is not reduced under our experimental conditions, but conversely, may be increased under low oxygen conditions (which would be expected to create an intracellular environment more rather than less favourable for viral replication). Our new data (updated Figure S2) show that the reduced HIV-1 replication observed under low oxygen conditions does not associate with an impaired T cell activation status.

Figure 2: Data shown suggests that low oxygen environment is unlikely to perturb HIV-1 entry to CD4 T cells by assessing the expression of the cell-surface receptors that are required for viral binding and internalization (i.e. CD4, CCR5 and CXCR4). However, T cells were cultured at 1% O₂ for only 24h while all experiment performed in Figure 1 were carried out (at 1% O₂) for 2 and 4 days. The authors should monitor the expression of these receptors after 2 and 4 days of hypoxia. Response: We have now measured CD4, CCR5 and CXCR4 expression on pre-activated CD4 T cells from ten different donors after 2 and 4 days of culture in 20% and 1% O₂. These new data are presented in Figure 2a-b and shows that low oxygen does not significantly affect CCR5 expression and has a modest enhancing effect on CD4 expression, causes a significant increase in surface CXCR4 expression levels. Since HIV requires both CD4 and one of the chemokine receptors to infect cells, the increase in CD4 and CXCR4 under 1% O₂ conditions would if anything be expected to facilitate infection of cells (particularly by CXCR4-utilising viruses) and is unlikely to explain the decrease in HIV-1 replication observed under these conditions.

In Fig.2 panel C, the authors concluded that the frequency of HIV integration events was comparable under 20% or 1% O₂ conditions. Yet, it is difficult to make such conclusion from only two individual donors. A statistically significant number of donors should be used (HIV integration events seems to be increased under 1% O₂). Response: We have included new data in Fig.2 from five additional donors to consolidate the results on strong stop, 1st strand and 2nd strand transfer as well as the integration. These data confirm our earlier observations that low oxygen has no significant effect on these viral parameters.

Figure 3: The authors evaluated the effect of 1% O₂ on HIV promotor activity using Jurkat 1G5 and HeLa TZM-bl cell lines -commonly used lines to monitor HIV-1 replication. Could the authors confirm that low oxygen conditions have no effect on the early steps of viral life cycle and HIV integrations events in these cells? Response: We performed additional experiments in Jurkat 1G5 and HeLa TZM-bl cells and confirmed that low oxygen conditions does not affect early steps of viral life cycle (strong stop, 1st and 2nd strand products) and HIV integrated copies in these cells, supported by new Supplementary Fig.S3.

Figure 4: Authors show that inhibition of HIF signaling restored HIV LTR promotor activity under low oxygen conditions. Could the authors confirm that the knock-down of HIF-1 α or HIF-2 α or treatment with NSC and 2-OG have no effect on cell viability? Response: We have incorporated new data showing cell viability is not compromised in either knockdown or treated Jurkat cells, this is presented in new Supplementary Fig.S4a.

Given that HIFs also play a fundamental role in coordinating cellular metabolism and function in T cells, such changes must be excluded. Also, these results should be confirmed in primary CD4 T cells. Response: We have edited the discussion to include new text on the role of HIFs in T cell function (page 7). In addition, we provide data showing that NSC and DMOG treatment of primary CD4 T cells phenocopies the results observed with Jurkat cells and this new data is presented in Fig.4. Additional data in Supplementary Fig.S5b shows that NSC has no impact on CD4 T cell viability while DMOG has a subtle effect.

In panel b, the reduction in HIF-2 α expression upon NSC treatment shown by western blot is not clear. Response: We agree with the reviewer and have provided a new HIF-2 α western blot image that is presented in updated Fig.4b.

Figure 5: To determine whether HIFs interact with the promotor in HIF infected Jurkat cells under 1% O₂, the authors performed ChIP experiment using HIF-1 α antibody. However, it is unclear why the authors used HIF-1 α antibody and not HIF-2 α since they showed in figure 4 that only HIF-2 α (and not HIF-1 α) knock-down was able to reverse the phenotype? Response: Please note we used an antibody specific for HIF1- β for our ChIP studies since this enables one to measure all HIF- α/β complexes bound to DNA and has been previously validated in ChIP-Seq studies.

Figure 6: The authors demonstrate that low oxygen conditions inhibit latent HIV reactivation using the established Jurkat latency model. It is unclear why the mean GFP expression (gMFI) is not statistically significant. Response: We'd like to clarify that low oxygen reduces both the frequency of reactivated cells (% GFP+) and their mean expression (gMFI) and this is represented in Fig.6c-d. I believe the reviewer is referring to data in Fig.6e that showed HIF inhibitor (NSC) inducing a significant increase in the frequency of activated GFP⁺ cells and a non-significant increase in GFP expression, that may reflect differences in GFP half-life under the experimental conditions used. Our revised Fig.6 includes data comparing the effect of NSC on TNF α or Romidepsin mediated J-Lat reactivation. Under low oxygen conditions, the combined NSC and TNF α treatment induced the greatest HIV reactivation. However, NSC did not increase the frequency of Romidepsin-reactivated cells, that is most likely explained via indirect effects of the HDAC inhibitor on HIF-transcriptional activity. Edits in revised MS (page 6, lines 9-14).

Also, where is the data showing that “under low oxygen conditions NSC-134754 and TNF α induced higher levels of HIV reactivation than Romidepsin” as the authors stated? Response: We thank the reviewer for the observation. We have included Romidepsin in the updated version of Figure 6e.

Minor points:

Typo “HIF-2 α ” line 4 in the discussion. Response: We have revised text to ensure a standardized nomenclature.

Together these comprehensive revisions have fully addressed all of the points raised by the reviewers and we look forward to a favourable decision on our manuscript.

REVIEWERS' COMMENTS:

Reviewer #3 (Remarks to the Author):

The authors have adequately addressed my concerns in their resubmission and comments. I recommend that this manuscript be accepted for publication.

Reviewer #4 (Remarks to the Author):

While addressing most of the points raised by the reviewers, and certainly improving the manuscript compared with the initial version, I am still concerned regarding the conditions of 1% used for the study, as opposed to the 3-5% which would better reflect the physiologically relevant oxygen tension.

The authors tried to address this point, albeit, not convincingly. The claim that most ex-vivo conditions use 1% oxygen does not address the need to perform the experiments in the correct level of oxygen for the lymphoid cells studied here. Since there is abundant literature concerning the differences in HIF regulation and function between the 1% and 4% conditions, authors either perform key experiments in 4%, or further modify the text to explain the expected differences while citing the relevant literature, provided such an option is acceptable to the editors.

Please find attached our response to the final comment from reviewer 4.

Reviewer 4: *I am still concerned regarding the conditions of 1% used for the study, as opposed to the 3-5% which would better reflect the physiologically relevant oxygen tension. The authors tried to address this point, albeit, not convincingly. The claim that most ex-vivo conditions use 1% oxygen does not address the need to perform the experiments in the correct level of oxygen for the lymphoid cells studied here. Since there is abundant literature concerning the differences in HIF regulation and function between the 1% and 4% conditions, authors either perform key experiments in 4%, or further modify the text to explain the expected differences while citing the relevant literature, provided such an option is acceptable to the editors.*

Response: Firstly we provide two additional references that measure oxygen tension in murine lymphoid organs, supporting our statement of 0.5-4.5% oxygen. We recognize the difficulties in modelling physiological oxygen tension of lymphoid organs in vivo and we provide new text to justify our choice of oxygen tension and in addition provide several new citations that also study T cell biology at 1% O₂ (Page 6, lines 33-35). We have also revised our text to state low oxygen rather than use the term 'physiological hypoxia' which may be open to mis-interpretation.

With best wishes,

Jane A McKeating,
Professor of Molecular Virology